# SARS-CoV-2 Causes Lung Inflammation through Metabolic Reprogramming and RAGE

**DOI:** 10.3390/v14050983

**Published:** 2022-05-06

**Authors:** Charles N. S. Allen, Maryline Santerre, Sterling P. Arjona, Lea J. Ghaleb, Muna Herzi, Megan D. Llewellyn, Natalia Shcherbik, Bassel E. Sawaya

**Affiliations:** 1Molecular Studies of Neurodegenerative Diseases Lab., FELS Cancer Institute for Personalized Medicine, Lewis Katz School of Medicine, Temple University, Philadelphia, PA 19140, USA; charlesnallen85@temple.edu (C.N.S.A.); tue86122@temple.edu (M.S.); sterling.arjona@temple.edu (S.P.A.); lea.ghaleb@temple.edu (L.J.G.); muna.herzi@temple.edu (M.H.); megan.llewellyn@temple.edu (M.D.L.); 2Department for Cell Biology and Neuroscience, School of Osteopathic Medicine, Rowan University, Stratford, NJ 08084, USA; shcherna@rowan.edu; 3Departments of Neurology, Lewis Katz School of Medicine, Temple University, Philadelphia, PA 19140, USA; 4Cancer and Cell Biology, Lewis Katz School of Medicine, Temple University, Philadelphia, PA 19140, USA; 5Neural Sciences, Lewis Katz School of Medicine, Temple University, Philadelphia, PA 19140, USA

**Keywords:** glycolysis, inflammation, metabolic reprogramming, mitochondria, RAGE, SARS-CoV-2, Tepp-46

## Abstract

Clinical studies indicate that patients infected with SARS-CoV-2 develop hyperinflammation, which correlates with increased mortality. The SARS-CoV-2/COVID-19-dependent inflammation is thought to occur via increased cytokine production and hyperactivity of RAGE in several cell types, a phenomenon observed for other disorders and diseases. Metabolic reprogramming has been shown to contribute to inflammation and is considered a hallmark of cancer, neurodegenerative diseases, and viral infections. Malfunctioning glycolysis, which normally aims to convert glucose into pyruvate, leads to the accumulation of advanced glycation end products (AGEs). Being aberrantly generated, AGEs then bind to their receptor, RAGE, and activate several pro-inflammatory genes, such as IL-1b and IL-6, thus, increasing hypoxia and inducing senescence. Using the lung epithelial cell (BEAS-2B) line, we demonstrated that SARS-CoV-2 proteins reprogram the cellular metabolism and increase pyruvate kinase muscle isoform 2 (PKM2). This deregulation promotes the accumulation of AGEs and senescence induction. We showed the ability of the PKM2 stabilizer, Tepp-46, to reverse the observed glycolysis changes/alterations and restore this essential metabolic process.

## 1. Introduction

COVID-19 is a disease caused by SARS-CoV-2 infection claiming the lives of more than 6 million across the globe since late 2019/early 2020 [1]. Despite the efforts to prevent this infection and treat the disease, an increasing number of studies have reported a variety of post-COVID-19 sequelae (known as *long COVID*), wherein chronic inflammation is one of the prevalent, persistent symptoms (Figure 1) [2]. Although it is known that SARS-CoV-2 reprograms the metabolism in several cell types causing chronic inflammation [3,4], the molecular and cellular mechanisms of this metabolic reprogramming are not fully understood.

Metabolic Reprogramming is a phenomenon observed in cancer, viral infection, and neurodegenerative diseases [5,6,7,8,9], which results in the rewiring of the cellular metabolism to support the increasing energy demand. Cancer cells and viruses use this mechanism to synthesize free nucleotides and amino acids necessary for their assembly and replication [10,11].

During normal glycolysis, glucose is metabolized into pyruvate, transferred into the mitochondria, where it is catabolized in the tricarboxylic acid cycle (TCA) and fuels oxidative phosphorylation to generate large amounts of ATP [12]. Viruses and tumors can alter cellular metabolism and increase pyruvate conversion into lactate in the presence of oxygen, a phenomenon commonly referred to as the Warburg effect [13]. In addition, metabolic reprogramming can lead to the oxidation of glucose-6P into fructose-1,6P to be used in the pentose phosphate pathway. This pathway can generate nucleotides to be used in viral genome replication [14]. Metabolic reprogramming can also result in increased fructose-1,6P conversion into glyceraldehyde-3P and increased dihydroxyacetone (DHAP), which then accumulates and promotes the formation of advanced glycation end-products (AGEs), which are toxic to the cell [15].

One process that has been shown to contribute to metabolic reprogramming is the last step of glycolysis where phosphoenolpyruvate (PEP) is converted into pyruvate. The conversion of PEP to pyruvate is controlled by pyruvate kinase muscle isoform (PKM) [16]. PKM has two distinct isoforms, 1 or 2, where PKM1 has constitutively active enzymatic activity and PKM2 enzymatic activity is allosterically controlled [17]. PKM2 is enzymatically active in the tetrameric form; however, it is inactive while it exists as a monomer and dimer. The phosphorylation of PKM2 on the tyrosine residue 105 (Tyr^105^) inhibits tetramer formation of PKM2, causing the enzymatic activity to cease [18]. Inactivation of PMK2 creates a roadblock in glycolysis through the decreased conversion of PEP into pyruvate [19]. The accumulation of glycolytic intermediates can lead to the accumulation of AGEs [20]. Excessive AGEs can bind and thus activate the receptor of advanced glycation end products (RAGE) [21].

Named for the ability to bind AGEs, RAGE belongs to the pro-inflammatory pattern recognition receptors (PRRs) that are relevant to many types of inflammatory diseases [22]. RAGE is expressed in vascular cells, immune cells, neuronal cells, cardiomyocytes, adipocytes, glomerular epithelial cells, podocytes, and lung epithelial cells [23]. In addition to AGEs, several other ligands can bind to RAGE, including S100 proteins, high mobility group box1 (HMGB1), lysophosphatidic acid (LPA), amyloid-beta (Aβ), islet amyloid polypeptide (IAPP), and macrophage 1-antigen (Mac-1) [24]. The binding of RAGE to its ligand activates mitogen-activated protein kinase (MAPK) and NF-κB and induces the production of various pro-inflammatory cytokines (e.g., IL-6) [25]. RAGE exists in two forms, membrane-bound RAGE (mRAGE) and soluble RAGE (sRAGE). mRAGE is composed of three domains (extracellular, hydrophobic transmembrane, and cytoplasmic), while sRAGE contains only the extracellular domain [26]. sRAGE, found in plasma, can bind to secreted AGEs and acts as a decoy for mRAGE resulting in decreased mRAGE signaling and ultimately clearance of AGEs [27,28].

Increased AGE accumulation can lead to an overwhelming of sRAGE capability and results in reduced AGE sequestration. This sRAGE exhaustion causes more AGEs to bind to mRAGE resulting in increased pro-inflammatory cytokine production [29]. Excessive inflammation is detrimental, especially for patients who suffer from underlying conditions, such as diabetes, arthritis, and Chronic Obstructive Pulmonary Disease (COPD) [30]. sRAGE plasma levels have been identified as a possible prognosticator for SARS-CoV-2 mortality [31]. Therefore, examination of the functional connection between acute inflammation and the AGE/RAGE pathway-dependent metabolic reprogramming represents a significant and clinically relevant task.

Here, using BEAS-2B cells transfected with SARS-CoV-2 plasmids, we showed the ability of SARS-CoV-2 proteins to induce the AGE/RAGE pathway, providing a molecular basis for the persistence of long COVID-19 sequelae that manifest as chronic inflammation rendering the AGE/RAGE pathway as a possible therapeutic target against SARS-CoV-2 mortality.

## 2. Materials and Methods

### 2.1. SARS-CoV-2 Plasmids

pLVX-EF1alpha-eGFP-2XStrep-IRES-Puro

pLVX-EF1alpha-IRES-Puro

pLVX-EF1alpha-nCoV2019-E-2xStrep-IRES-Puro

pLVX-EF1alpha-nCoV2019-M-2xStrep-IRES-Puro

pLVX-EF1alpha-nCoV2019-N-2xStrep-IRES-Puro

pLVX-EF1alpha-nCoV2019-nsp1-2xStrep-IRES-Puro

pLVX-EF1alpha-nCoV2019-nsp2-2xStrep-IRES-Puro

pLVX-EF1alpha-nCoV2019-nsp4-2xStrep-IRES-Puro

pLVX-EF1alpha-nCoV2019-nsp5-2xStrep-IRES-Puro

pLVX-EF1alpha-nCoV2019-nsp7-2xStrep-IRES-Puro

pLVX-EF1alpha-nCoV2019-nsp8-2xStrep-IRES-Puro

pLVX-EF1alpha-nCoV2019-nsp9-2xStrep-IRES-Puro

pLVX-EF1alpha-nCoV2019-nsp10-2xStrep-IRES-Puro

pLVX-EF1alpha-nCoV2019-nsp11-2xStrep-IRES-Puro

pLVX-EF1alpha-nCoV2019-nsp12-2xStrep-IRES-Puro

pLVX-EF1alpha-nCoV2019-nsp13-2xStrep-IRES-Puro

pLVX-EF1alpha-nCoV2019-nsp15-2xStrep-IRES-Puro

pLVX-EF1alpha-nCoV2019-orf3a-2xStrep-IRES-Puro

pLVX-EF1alpha-nCoV2019-orf6-2xStrep-IRES-Puro

pLVX-EF1alpha-nCoV2019-orf7a-2xStrep-IRES-Puro

pLVX-EF1alpha-nCoV2019-orf8-2xStrep-IRES-Puro

pLVX-EF1alpha-nCoV2019-orf9b-2xStrep-IRES-Puro

pLVX-EF1alpha-nCoV2019-orf10-2xStrep-IRES-Puro

pLVX-EF1alpha-IRES-Puro_2xStrep-nCoV2019-nsp14

pLVX-EF1alpha-IRES-Puro_2xStrep-nCoV2019-orf3b

pLVX-EF1alpha-IRES-Puro_2xStrep-nCoV2019-orf7b

pLVX-EF1alpha-IRES-Puro_2xStrep-nCoV2019-orf9c

pGBW-m4134157 (nsp6 plasmid) was a gift from Ginkgo Bioworks & Benjie Chen (Addgene plasmid # 152062; http://n2t.net/addgene:152062 accessed on 21 July 2020; RRID:Addgene_152062). All other plasmids were a gift from the Nevan Krogan Lab, UCSF, San Francisco, CA, USA.

### 2.2. Cell Culture and Transfection

Lungs cells (BEAS-2B) were maintained at 37 °C in Dulbecco’s modified Eagle’s medium (DMEM) (Cellgro, Manassas, VA, USA) with 2% fetal bovine serum (FBS), 100 units/mL of penicillin, 100 μg/mL of streptomycin. The cells were transfected with 1 μg of the empty vector or the SARS-CoV-2-expression plasmids (described above) [32] using Lipofectamine 3000 (Invitrogen, Carlsbad, CA, USA) as recommended by the manufacturer. Briefly, 0.75 µL of Lipofectamine 3000 Reagent was diluted into 25 µL of Opti-MEM medium and 1 µg of plasmids were diluted into 50 µL of Opti-MEM medium with 10 µL of P3000 Reagent added. The diluted DNA was then added to the diluted Lipofectamine 3000 Reagent and allowed to incubate for 15 min at room temperature. After incubation, the mixture was then added to the cells.

### 2.3. Chemical Reagents

Tepp-46 (ML-265) is a selective pyruvate kinase M2 (PKM2) tetramer stabilizer [33]. Transfected BEAS-2B cells were treated with 10 nM of Tepp-46 (purchased from VWR, Radnor, PA, USA) for four hours before downstream assays and measurements were conducted.

### 2.4. Western Blot Assay

Proteins were extracted from cell pellets using a radioimmunoprecipitation assay (RIPA) lysis buffer (25 mM Tris-HCl pH 7.6, 150 mM NaCl, 1% Triton x-100, 0.1% SDS, and 1x protease inhibitor cocktail). Estimation of protein concentrations was performed using bicinchoninic acid (BCA) assay (Thermo Fisher Scientific, Waltham, MA, USA). Protein samples are prepared in Laemmli buffer, and Western blot was performed using 20 μg of protein per well using polyacrylamide gels. The gels were then transferred to nitrocellulose membranes, rinsed for 10 min 3 times for 10 min each time, and blocked in 10% milk for 20 min. Primary antibodies were diluted in 5% milk and allowed to incubate at 4 °C overnight. The next day, the membrane was rinsed again in TBST for 10 min 3 times and secondary antibodies were added, diluted in 5% milk, for 1 h. Imaging was performed using a Thermo Fisher iBright imaging system. Antibodies and concentrations used are as follows: PKM2 (1/1000; D78A4), PTBP1 (1/1000; 12582-1-Ap), NF-κB-p65 (1/1000; C-20; sc-372), secondary anti-mouse (1/5000; 71045-3), secondary anti-rabbit (1/10000; AP187P) [Sigma-Aldrich, St. Louis, MO, USA], and H3 (0.5 μg/mL; A01502) [GenScript, Piscataway, NJ, USA]. ImageJ was used to determine the densitometry ratios of the bands. Western blot loading was normalized to the loading control, H3.

### 2.5. RNA Extraction

Global RNA was extracted using the Absolutely RNA Miniprep kit (Agilent Technologies, Santa Clara, CA, USA, 400800). The purity and concentration of extracted RNA were determined using a NanoDrop 2000 spectrophotometer (Thermo Scientific, Waltham, MA, USA).

### 2.6. qPCR Assay

cDNA was synthesized from isolated RNA using SuperScript IV VILO cDNA Master Mix with ezDNase (Invitrogen, Waltham, MA, USA, 11766050). Briefly, for each reaction, 4 µL of SuperScript IV VILO master mix was mixed with 2.5 µg of RNA and then diluted with water to reach 20 µL. The samples were then incubated for 10 min at 25 °C, then 10 min at 50 °C, and finally for 5 min at 85 °C. The following primers (purchased from IDT) were used: HIF-1α: (F) 5′-gaacgtcgaaaagaaaagtctcg-3′; (R) 5′-ccttatcaagatgcg aactcaca-3′. PTBP1: (F) 5′-aatgacaagagccgtgactac-3′; (R) 5′-ggaaccagctcctgcatac-3′. PKM1: (F) 5′-cgagcctcaagtcactccac-3′; (R) 5′-acgacgtcaccccggtattagc-3′. PKM2: (F) 5′-attatttgagga actccgccgcct-3′; (R) 5′-attccgggtcacagcaatgatgg-3′. IL-1β: (F) 5′-tacctgtcctgcgtgttgaa-3′; (R) 5′-tctttgggtaatttttgggatct-3′. IL8: (F) 5′-agacagcagagcacacaagc-3′; (R) 5′-atggttccttccggtggt-3′. RAGE: (F) 5′-gggcagtagtaggtgctcaa-3′; (R) 5′-tccggcctgtgttcagtttc-3′. GAPDH: (F) 5′-caaggctgagaacgggaag-3′; (R) 5′-tgaagacgccagtggactc-3′. qPCR was performed using FastStart Universal SYBR Green (Roche, Basel, Switzerland, 04913914001), and all reactions were performed according to the manufacturer’s instructions. Briefly, 25 µL of FastStart Universal SYBR Green Master mix was mixed with 300 nM of both forward and reverse primers and water to bring reaction to 45 µL, then 50 ng (5 µL) of cDNA was added. The reaction was performed on a ThermoFisher thermal cycler. The relative quantitation of mRNA was performed using the comparative ∆∆Ct method. The results were then compared to those of the control group and endogenous control, GAPDH.

### 2.7. Seahorse Mito Stress Test

After transfection with SARS-CoV-2 plasmids, changes in oxygen consumption were analyzed using the XFe96 Seahorse Analyzer (Agilent Technologies, Santa Clara, CA, USA). BEAS-2B cells were split onto an XFe96-well microplate and were cultured overnight. On the day of the experiment, the growth medium was replaced with an XF assay medium containing DMEM supplemented with 10 mM glucose, 4 mM l-glutamine, and 2 mM sodium pyruvate. The cells were then placed in an incubator set at 37 °C without CO_2_. Extracellular acidification rate (ECAR) and oxygen consumption rate (OCR) measurements were made following the manufacturer’s recommendations and took place at basal conditions and in response to 1 μM Oligomycin, 1.5 μM FCCP, and 1 μM rotenone/1 μM antimycin A [34].

### 2.8. Pyruvate Assay

Pyruvate production was conducted using a fluorometric Pyruvate Assay Kit (Cayman Chemical, Ann Arbor, MI, USA, C789C04). BEAS-2B cells were transfected with SARS-CoV-2 plasmids as above and treated with Tepp-46 for 8 h. After 8 h, the cells were collected, and the assay was carried out according to the manufacturer’s protocol. Briefly, cell pellets were collected and rinsed in 1 mL of PBS and centrifuged at 10,000× *g* for 5 min, and the supernatant was removed from the cell pellet. Then, 0.5 mL of 0.25 M MPA was added to the cell pellet, vortexed for 30 s, and placed on ice for 5 min. After 5 min, the samples were centrifuged at 10,000× *g* for 5 min, and the supernatant was removed. To the supernatant, 25 µL of potassium carbonate was added. The samples were then centrifuged again at 10,000× *g* for 5 min, and then the supernatant was diluted 1:2 with Assay Buffer. Then, 20 µL of the samples were added to a 96-well plate along with 50 µL of Assay Buffer, 50 µL of Cofactor Mixture, and 10 µL of Fluorometric Detector. The reaction was then initiated by adding 20 µL of Enzyme Mixture to each well and allowed to incubate for 20 min at room temperature. The plate was read on a plate reader using an excitation wavelength between 530 and 540 nm and an emission wavelength between 585 and 595 nm. Pyruvate results were normalized to each sample’s cell count. Pyruvate concentration was calculated by comparison to the standard curve and using the formula pyruvate (µM) = [(CF-y-intercept)/slope] × 2 × sample dilution.

### 2.9. Human Advanced Glycation End-Products (AGEs) ELISA

Advanced glycation end-product (AGE) concentrations were determined using the Immunotag Human AGEs ELISA kit (G biosciences, St. Louis, MO, USA, cat# IT1931). The ELISA is reactive to human AGE including Nε-Carboxy-methyl-lysine (CML) and Nε-carboxy-ethyl-lysine (CEL). The ELISA assay was performed according to the manufacturer’s protocol. Briefly, BEAS-2B cells were transfected with SARS-CoV-2 plasmids and treated with or without Tepp-46 for 8 h. The cells were then collected and centrifuged (800× *g*) for 5 min. The supernatant was then removed, and 100 μL of RIPA buffer with added protease inhibitors was then added to the cell pellet and rotated at 4 °C for an hour. The ELISA plate was prepared following the manufacturer’s protocol. The samples were diluted at 1:5 using the provided Sample Dilution buffer. The diluted samples were added to the ELISA plate and incubated at 37 °C for 90 min. After incubation, the provided biotin detection antibody was added to the plate. The plate was then incubated again for 60 min at 37 °C. After 60 min, the HRP-Streptavidin Conjugate was added to the plate and incubated again at 37 °C for 30 min. The provided TMB Substrate was then added to the plate and incubated in the dark at 37 °C for 15 min. After determining that the reaction achieved optimal levels by using the color of the standard curve wells, the Stop Solution was added to end the reaction. The plate was then read at 450 nm. Concentrations were derived using the standard curve, then the samples were normalized to dilution factor and cell count.

### 2.10. Cell Senescence Staining

Senescence was evaluated using a β-galactosidase cell staining kit (Cell Signaling, Danvers, MA, USA) as recommended by the manufacturer. Briefly, BEAS-2B cells were cultured in 3.5 cm dishes were rinsed, fixed with 1 × fixative solution for 15 min, and washed twice with PBS. β-galactosidase staining solution (930 μL 1 × staining solution with 10 μL supplement A, 10 μL supplement B, and 50 μL 20 mg/mL X-gal in DMF) was added, and the cells were incubated at 37 °C without CO_2_ overnight. Images were taken with a Nikon LED microscope the following day, as previously described [35]. Experiments were repeated two times.

### 2.11. Immunofluorescence Staining and Confocal Microscopy

BEAS-2B cells transfected with SARS-CoV-2 plasmids were plated on glass coverslips and allowed to grow overnight. The cells were then fixed in 4% formaldehyde in PBS solution for 20 min. The cells were then washed in PBS, and 0.2% Triton x100 was used to permeabilize the cells. Afterward, the cells were incubated with anti-GOLGA/GM130 antibody (1/350, Proteintech, Rosemont, IL, USA, 11308-1-AP) overnight at 4 °C. Cells were washed twice with 0.1% PBST and then incubated in Alex Fluor 488-conjugated AffiniPure Goat Anti-Rabbit IgG(H + L) for at least 1 h. Images of fluorescent staining were taken on a confocal microscope (Leica, Wetzlar, Germany). The fluorescence intensity of GOLGA/GM130 was determined by background subtraction and using a fixed threshold in ImageJ. The experiment was performed following the protocol previously described [36].

### 2.12. Statistical Analysis

All statistical analyses were performed using either one-way analysis of variance (ANOVA) or a Student’s t-test. Data are displayed as the mean with ±1 standard deviation (S.D.). Results were identified as statistically significant if *p* < 0.05 (marked in the figures as * *p* < 0.05; ** *p* < 0.01; *** *p* < 0.001; **** *p* < 0.0001 where needed). Data were plotted using GraphPad Prism version 7.0.

## 3. Results

### 3.1. SARS-CoV-2 Proteins Disrupt the OCR

First, we sought to determine if SARS-CoV-2 proteins spike, envelope (Env), or all SARS-CoV-2 proteins except the membrane protein (All), disrupt mitochondrial oxygen consumption. We measured the oxygen consumption rate (OCR) (Figure 2A) and the extracellular acidification rate (ECAR) (Figure 2B) using a Seahorse Mito Stress Test. The spike protein but not the envelope protein was able to disrupt the oxygen consumption rate when compared to the control BEAS-2B cells (Figure 2A). In addition, the combination of all the proteins also led to the disruption of mitochondrial-associated oxygen consumption (yellow line) (Figure 2A). Furthermore, the BEAS-2B cells expressing all the SARS-CoV-2 proteins, and the cells expressing the spike protein alone decreased the OCR associated with maximal respiration. The decrease in OCR is representative of decreased ability of cellular respiration through the electron transport chain and ultimately decreased oxidative phosphorylation. Decreases in OCR are indicative of mitochondrial damage and reduced substrate availability to enter the TCA cycle [37]. In addition, the cells transfected with the combination of all the SARS-CoV-2 proteins and the spike protein also result in a decrease in the spare respiration capacity, when compared to the control untreated cells (Figure 2C,D). The spare respiration capacity is the difference between the capacities at maximum respiration and basal respiration. The decrease in spare respiration capacity also further indicates the lack of substrates to enter the TCA cycle [38].

### 3.2. SARS-CoV-2 Increases PKM2 over PKM1

Decreased OCR indicates altered glycolysis and reduced pyruvate production [39]. Based on our observations and the literature, many viruses can alter the glycolysis pathway at the last step of glycolysis [40]. The last step is the conversion of phosphoenolpyruvate (PEP) into pyruvate and is performed by the pyruvate kinase enzyme, which has four isoforms including kinase muscle isoforms 1 and 2 (PKM1 and PKM2). PKM1 and PKM2 are the dominant isoforms of PKM that are found in most tissues [41]. The polypyrimidine tract-binding protein 1 (PTBP1) is important for splicing PKM into the PKM2 isoform over PKM1 [42,43]. Therefore, using BEAS-2B cells which produce a combination of all the SARS-CoV-2 DNA except the membrane protein, we determined the expression levels of PTBP1 and PKM2 proteins through a Western blot assay using anti-PTBP1 and -PKM2 antibodies. The production of SARS-CoV-2 proteins led to an increase in the expression of PTBP1 and PKM2 proteins (Figure 3A,B). The SARS-CoV-2 Env protein failed to affect the mitochondrial oxygen consumption rate (Figure 2A); however, it did upregulate the expression of both PTBP1 and PKM2 proteins (Figure 3A,B).

In addition, mRNA was extracted and subjected to qPCR to measure the expression levels of the indicated factors. As shown, transfection of the cells with SARS-CoV-2 plasmids led to increased mRNA expression of PTBP1 and PKM2 (Figure 3C,D). The SARS-CoV-2 proteins also decreased the expression of PKM1 mRNA (Figure 3E). The increase in PTBP1 and PKM2 due to the SARS-CoV-2 proteins indicates viral-associated metabolic reprogramming resulting in increased glycolysis that stalls at the final step resulting in reduced pyruvate production but increased glycolytic intermediates [44].

### 3.3. Tepp-46 Reverses the Effect of the SARS-CoV-2 Protein

The increase in PTBP1 and PKM2 due to the SARS-CoV-2 proteins indicates viral-associated metabolic reprogramming resulting in increased glycolysis that stalls at the final step resulting in reduced pyruvate production but increased glycolytic intermediates.

Since PKM2 enzymatic activity depends on what confirmation it is in, dimer or tetramer, depending on allosteric controls, we set out to determine if PKM2 confirmation plays a role in the metabolic reprogramming of BEAS cells due to SARS-CoV-2 proteins. To do this, we used a known pharmaceutical PKM2 tetramer stabilizer, Tepp-46. Hence, we measured the concentration of intercellular pyruvate in BEAS-2B cells producing SARS-CoV-2 proteins as indicated. As expected, the production of SARS-CoV-2 proteins led to decreased concentration of internal pyruvate (Figure 4A), indicating that the PEP conversion into pyruvate is altered due to the increased expression of PKM2. The decreased conversion of PEP into Pyruvate, and the increase in PKM2 expression, suggest that PKM2 is in the non-enzymatic dimer configuration and is causing increased glycolysis and the accumulation of glycolytic intermediates [44,45].

We have previously shown in neurons that have undergone viral metabolic reprogramming due to HIV-1 gp120 that Tepp-46 can reverse the increased glycolysis and decrease the TCA cycle due to the increase in PKM2 dimers [45]. Furthermore, the decreased conversion of PEP into Pyruvate along with the increase in PKM2 expression suggests that PKM2 is in the non-enzymatic dimer configuration.

### 3.4. SARS-CoV-2 Viral Proteins Result in Increased Advanced Glycation End-Products (AGEs)

Recently, it has been shown that metabolic reprogramming due to viruses’ infection can result in increased glycolysis and decreased TCA cycle and increases the production of advanced glycation end-products (AGEs) [46]. AGES can be formed spontaneously from increased glycolytic intermediates and oxidative stress [47]. We have previously shown that the increase in AGEs can be attributed to the increase in PKM2 in neurons that have been exposed to HIV-1 viral protein gp120 [44]. To explore if SARS-CoV-2 proteins have a similar increase in AGEs, we transfected BEAS cells with the complete array of viral plasmids except for the membrane plasmid. In the cells transfected with the SARS-CoV-2 plasmids, we observed an increase in the concentration of AGEs, specifically Nε-Carboxy-methyl-lysine (CML) and Nε-carboxy-ethyl-lysine (CEL) (Figure 4B). CML and CEL both have previously been shown to bind to RAGE and induce inflammation and have been linked to many metabolic reprograming related diseases [48]. In the BEAS cells transfected with the combination of SARS-CoV-2 plasmids, we also show that the addition of Tepp-46 reverses AGEs accumulation, indicating that Tepp-46 favors the formation of the tetrameric form of PKM2 and reduces the glycolytic intermediates required to spontaneously produce AGEs resulting in reduced AGE accumulation (Figure 4B).

AGEs can bind to mRAGE and create a positive feedback loop resulting in increased RAGE transcription [48]. The forward feedback loop would result in increased RAGE production and would contribute to increased RAGE signaling resulting in compounded increased inflammation. To validate this, we first examined whether the addition of viral plasmids activates this pathway. BEAS-2B cells were transfected with SARS-CoV-2 plasmids for 48 h, then subjected to Western blot analysis. The transfection of BEAS-2B cells with the SARS-CoV-2 plasmids increases the expression level of RAGE RNA (Figure 4C). Treatment of the cells with Tepp-46 modestly reversed the effect of SARS-CoV-2 proteins on RAGE RNA (Figure 4C).

Finally, HIF-1a has been shown to increase PTBP1 expression and increase the splicing of PKM2 over PKM1 [45]. We observe that the SARS-CoV-2 proteins increase the expression of HIF-1a, and we also show that Tepp-46 fails to modify the expression levels of HIF-1a (Figure 4E). Since HIF-1a is upstream of PKM2 conformation and not affected by the enzymatic activity of PKM2, it makes sense that Tepp-46 did not affect these factors.

### 3.5. SARS-CoV-2 Proteins Induce Cytokines and Promote Cell Senescence

AGEs can bind to mRAGE, leading to the production of several pro-inflammatory cytokines [49]. The binding of RAGE to its ligand AGEs activates the NF-kB pathway leading to the activation of several pro-cytokines [50]. To validate this hypothesis, we first examined whether the addition of the viral proteins activates this pathway. BEAS-2B cells were transfected with several SARS-CoV-2 plasmids for 48 h, then subjected to Western blot analysis using an anti-p65 antibody (p65 is one of the five components that form the NF-κB). As shown in Figure 5A. the addition of SARS-CoV-2 proteins increases the expression levels of p65 in these cells compared to the untransfected control.

Next, we measured the expression levels of IL-1β and IL-8 mRNA in BEAS-2B cells transfected for 44 h with SARS-CoV-2 plasmids in the presence and absence of Tepp-46. As shown in Figure 5B,C, expression levels of IL-1β and IL-8 increased in SARS-CoV-2-treated proteins and decreased with the addition of Tepp-46. These results confirm the hypothesis that SARS-CoV-2 depends on PKM2 to induce pro-inflammatory cytokines through the activation of RAGE and the accumulation of AGEs. Stabilizing PKM2 reverses this effect and lowers the inflammation.

We also sought to determine whether the induction of pro-inflammatory cytokines can promote cell senescence. BEAS-2B cells were transfected with SARS-CoV-2 proteins for 48 h, and then a beta-galactosidase assay was performed. Indeed, these proteins induce cell senescence, as displayed in Figure 5D.

Senescence is known to enhance the secretion of proteins that promote inflammation, and since it has been shown that the Golgi complex regulates this secretion [50], we examined the status of Golgi in these cells. Moreover, the structure of the Golgi complex is altered in senescent cells [50]. As shown in Figure 5E, the addition of SARS-CoV-2 proteins causes the fragmentation of the Golgi apparatus. These results gave the rationale to conclude that SARS-CoV-2/COVID-19 causes long-term sequelae, especially persistent inflammation through the activation of the AGE/RAGE pathway.

## 4. Discussion

Even after its clearance, SARS-CoV-2/COVID-19 sequelae persist for several months and could negatively affect the patients’ daily life (Figure 1). Seventy different systemic organ-specific disorders were identified to be associated with post-COVID-19 symptoms [51]. The mechanisms involved are not fully understood; however, most clinicians and researchers agree that inflammation is the number one reason for these long-term sequelae followed by the induction of hypoxia [52,53,54,55]. Both inflammation and hypoxia are the results of metabolic reprogramming (MR) caused by the virus and persist in patients, especially those with underlying diseases.

The molecular and cellular mechanisms leading to chronic inflammation remain unclear, and the cellular factor(s) implicated in the persistence of inflammation are also unknown. We hypothesize that SARS-CoV-2 mimics other viruses in reprogramming the metabolism resulting in chronic inflammation [55]. Several labs described that SARS-CoV-2 reprograms the metabolism to aid in viral replication [56,57,58].

We found that the virus changes the metabolism and induces pro-inflammatory cytokines through a pathway involving AGEs and the RAGE receptor. We also demonstrated that the viral proteins alter the glycolysis pathway resulting in increased glycolytic intermediates and decreased pyruvate. This phenomenon has a precedent where viruses use this pathway to replicate [46]. Studies showed that HCV deregulates the mitochondria, promotes HIF-1α, and activates the glycolysis pathway [59]. Additionally, cells infected with the human cytomegalovirus (CMV) showed increased glycolysis and disrupted TCA [60]. Similar events were observed in B-cells infected with EBV [61] or T-cells infected with HTLV-1 [62]. We demonstrated that HIV-1 gp120 protein causes comparative symptoms in neurons [44]. We also showed that metabolic reprogramming is composed of seven hallmarks, and any disruption of these hallmarks can lead to the accumulation of RAGEs and chronic inflammation; however, the role of RAGE remains understudied [46].

RAGE was described to be elevated in SARS-CoV-2-infected patients and to play a role in chronic infection associated with COVD-19 [63]. RAGE was also classified among the 40 markers pointing to damaged organs in SARS-CoV-2-infected patients [64]. We found that the addition of Tepp-46 reverses the effect of the viral proteins. These results confirm that the viral proteins deregulate the glycolysis pathway and result in the formation of the dimeric form of PKM2. Monomeric or dimeric PKM2 prevents the conversion of PEP into pyruvate and alters the TCA cycle. The persistence of these symptoms increases the inflammation and may cause cell death.

Hence, to escape death, the cells stop dividing and enter a passive state, senescence, where additional factors involved in inflammation are secreted. These factors are secreted either by the mitochondria and known as senescence-associated secreted phenotype (SASP) or through the activation of the AGEs/RAGE pathway. This causes the fragmentation of the Golgi apparatus because the Golgi regulates the secretion of SASP [50]. The data presented in Figure 5 confirmed this hypothesis.

Understanding the mechanisms leading to RAGEs’ activation and chronic inflammation (Figure 6) permit the identification of new therapies to prevent long-term COVID-19 sequelae [65]. The AGE/RAGE pathway presents a possible pharmaceutical target to reduce inflammation in general and mortality due to COVID-19.

## Figures and Tables

**Figure 1 viruses-14-00983-f001:**
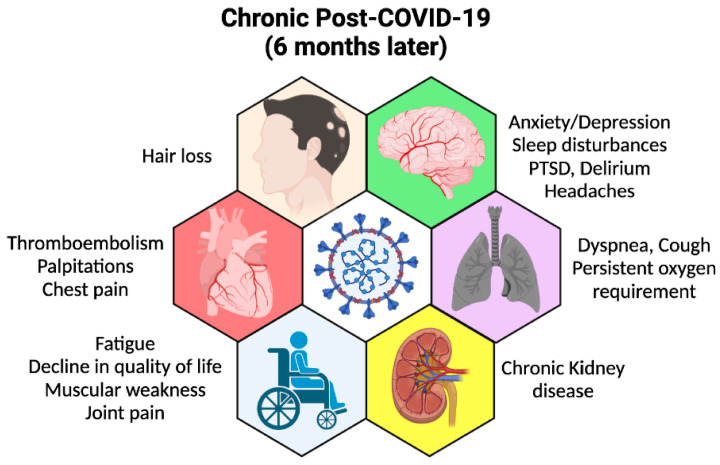
Long COVID-19 or post-acute sequelae. Schematic representation of persistent symptoms post-acute COVID-19. The figure was created using the BioRender.com website accessed on 15 March 2022.

**Figure 2 viruses-14-00983-f002:**
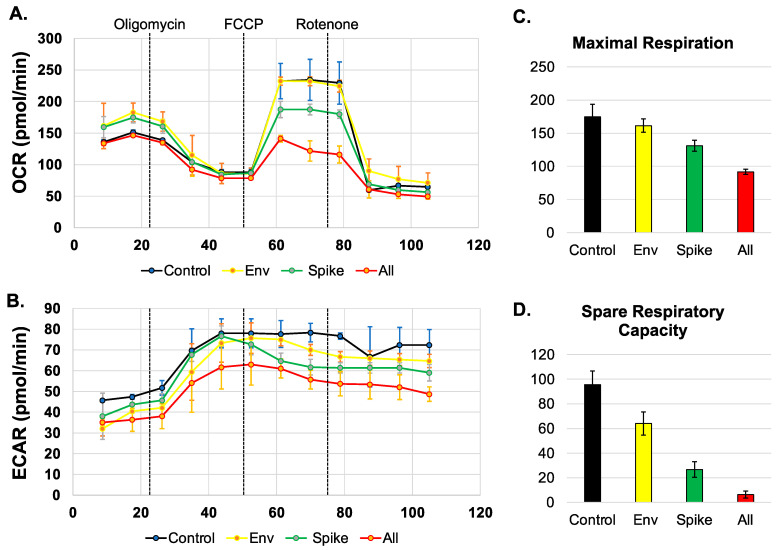
SARS-CoV-2 proteins alter mitochondrial maximal respiration. (**A**,**B**). Measurement of oxygen consumption rate (OCR) and extracellular acidification rate (ECAR) in BEAS-2B cells untreated or transfected with SARS-CoV-2 proteins for 48 h using a Seahorse XFe96 analyzer and a Mito stress test kit from Seahorse. Oligomycin, FCCP, and rotenone/antimycin (arrows) were added at 23, 50, and 75 min, respectively. (**C**,**D**). Graphs displaying the maximal respiration and spare capacity measured (pmol/min).

**Figure 3 viruses-14-00983-f003:**
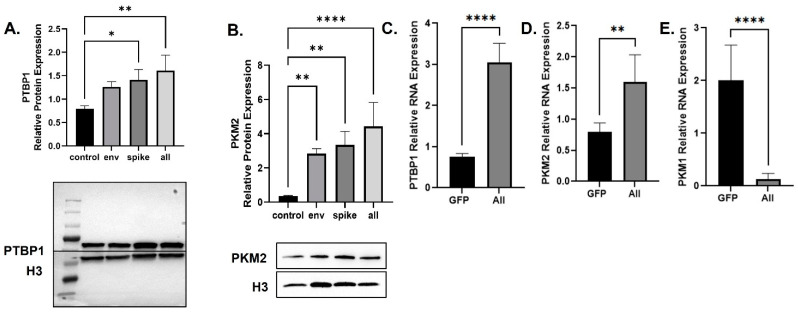
Changes in Proteins Responsible for Metabolic Reprogramming. Expression of PTBP1 (**A**) and PKM2 (**B**) proteins isolated from untreated or BEAS-2B transfected as indicated. Quantification of the relative protein levels was determined from the band intensity using ImageJ software and normalized relative to the H3. PTBP (**C**), PKM1 (**D**), and PKM2 (**E**) mRNA expression was obtained by qPCR using BEAS-2B transfected with all SARS-CoV-2 plasmids except for the membrane. Bar graphs represent the means ± S.D. of at least two independent experiments. Data represent the mean ± S.D. Results were judged statistically significant by ANOVA (* *p* < 0.05; ** *p* < 0.01; **** *p* < 0.0001).

**Figure 4 viruses-14-00983-f004:**
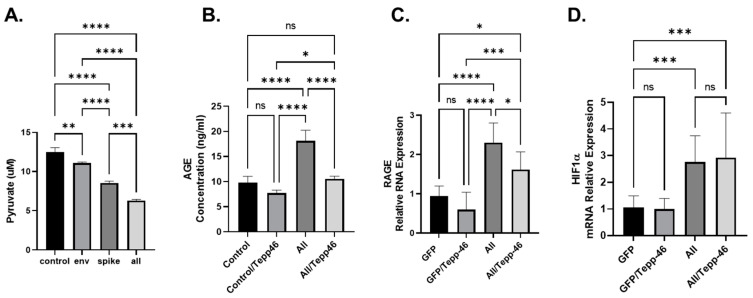
(**A**). Concentration of intercellular pyruvate in BEAS-2B cells transfected with SARS-CoV-2 plasmids as indicated (Envelope (Env.), Spike, or All (all SARS-CoV-2 plasmids except for the membrane). (**B**). Cellular concentration of AGEs (ng/mL) was measured by ELISA using BEAS-2B transfected with all SARS-CoV-2 plasmids except for the membrane. (**C**,**D**). mRNA expression of RAGE and HIF-1α. mRNA measured by qPCR using BEAS-2B transfected with all SARS-CoV-2 plasmids except for the membrane. Results were judged statistically significant by ANOVA (ns = not significant; * *p* < 0.05; ** *p* < 0.01; *** *p* < 0.001; **** *p* < 0.0001).

**Figure 5 viruses-14-00983-f005:**
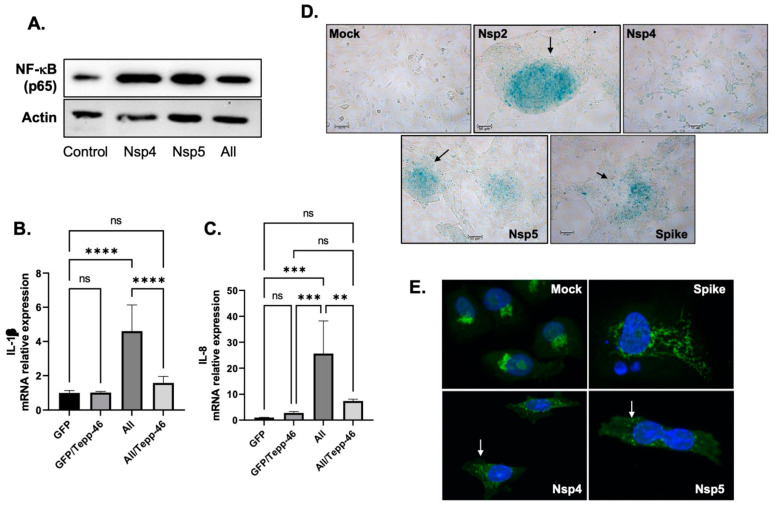
Persistence of Inflammation. (**A**). Expression of p65 subunit of NF-kB proteins isolated from untreated or BEAS-2B transfected as indicated by Western blot analysis. (**B**,**C**). mRNA expression of IL-1β and IL-8 as obtained by qPCR using BEAS-2B transfected then treated with 10 nM of Tepp-46 as indicated. Results were judged statistically significant by ANOVA (ns = not significant; ** *p* < 0.01; *** *p* < 0.001; **** *p* < 0.0001). (**D**), Nsp2, Nsp4 (modest), Nsp5, and the Spike proteins promote senescence in BEAS-2B as obtained using β-gal assay compared to the Mock untransfected control. (**E**). The addition of Nsp4 and NNsp5 to BEAS-2B causes a strong Golgi apparatus fragmentation compared to the Mock or the spike alone.

**Figure 6 viruses-14-00983-f006:**
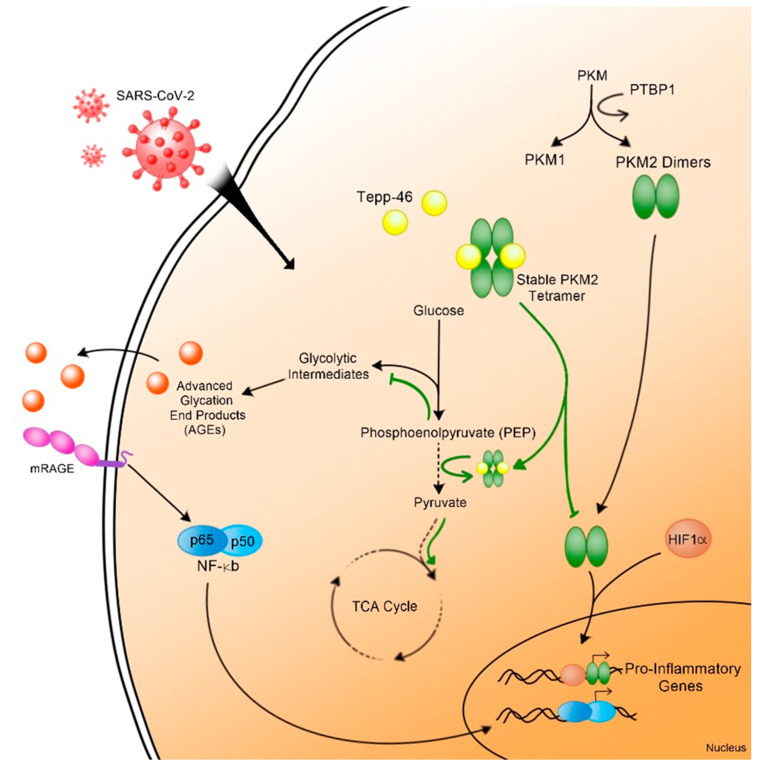
Schematic representation of metabolic reprograming leading to RAGE signaling and inflammation pathways in post-acute COVID-19 (black arrows). PKM2 stabilizer, Tepp-46, reverses this metabolic reprogramming and inhibits RAGE signaling (green arrows).

## Data Availability

All processed data are included in this manuscript. Raw data, further information, or reagents contained within the manuscript are available upon request from the corresponding author, Bassel E Sawaya, sawaya@temple.edu.

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
