# Peer review of "SARS-CoV-2 Causes Lung Inflammation through Metabolic Reprogramming and RAGE"

_viruses, 2022, doi:10.3390/v14050983_

Round 1

Reviewer 1 Report

The manuscript is an original work merits publication.

Minor comment:

AGEs concentrations were determined using AGEs ELISA kit. As known, there are many AGEs products, can the authors add some informations on what kind of products were measured?

Author Response

We thank the reviewers for their constructive comments.

We did the requested info in the revised version.

Reviewer 2 Report

The article written by Allen CNS et al, is well designed and well written. It prove using some molecular technics that infection by the virus SARS-CoV-2 induce a cascade of metabolic programmation witch finally lead to inflammation and the post COVID-19 symptoms. 

Minor remark:

  • Authors should add more informations in the molecular methods description in MM section...The design of the work is original and more precisions to add in methods section is important for readers.

Author Response

We thank the reviewers for their constructive comments.

We did the requested info in the revised version.

More details about the Methods are now added.

We replaced Fig 6 with a better model.

Round 2

Reviewer 1 Report

Manuscript with this version merits publication